# Glucose-Modified Zein Nanoparticles Enhance Oral Delivery of Docetaxel

**DOI:** 10.3390/pharmaceutics14071361

**Published:** 2022-06-27

**Authors:** Yabing Xing, Xiao Li, Weiwei Cui, Meng Xue, Yanan Quan, Xinhong Guo

**Affiliations:** 1Department of Pharmacy, Children’s Hospital Affiliated to Zhengzhou University, Zhengzhou 450018, China; xyb06@live.com; 2School of Pharmaceutical Sciences, Zhengzhou University, Zhengzhou 450001, China; g13569833579@gs.zzu.edu.cn (X.L.); 202022342015534@gs.zzu.edu.cn (W.C.); xm@gs.zzu.edu.cn (M.X.); quanyanan@stu.zzu.edu.cn (Y.Q.); 3Key Laboratory of Advanced Pharmaceutical Technology, Ministry of Education of China, Zhengzhou 450001, China

**Keywords:** glucose transporter, zein nanoparticles, docetaxel, anti-tumor effect, nanocarriers

## Abstract

Based on glucose (G) transporters (GLUTs), structuring nanoparticles with G as a target are an effective strategy to enhance oral bioavailability and anti-tumor effects of drugs. A novel drug delivery system using G-modified zein (GZ) nanoparticles loaded with docetaxel (DTX) (DTX-GNPs) was prepared and characterized *in vitro* and *in vivo* via assessment of cellular uptake, absorption site, pharmacokinetics, *ex vivo* distribution, and anti-tumor effects. The DTX-GNPs were approximately 120 nm in size. Compared with DTX-NPs, G modification significantly enhanced cellular uptake of DTX-GNPs by 1.22 times in CaCo-2 cells, which was related to GLUT mediation and the enhancement of endocytosis pathways *via* clathrin, micropinocytosis, and caveolin. Compared to DTX-NPs, G modification significantly enhanced DTX-NP absorption in the jejunum and ileum, delayed plasma concentration peak time, prolonged the average residence time *in vivo*, and increased oral bioavailability (from 43.82% to 96.04%). Cellular uptake and oral bioavailability of DTX were significantly affected by the G modification ratio. Compared with DTX-NPs, G modification significantly reduced drug distribution in the liver, lungs, and kidneys and increased tumor distribution and tumor growth inhibition rate without obvious systemic toxicity. This study demonstrated the potential of GZ-NPs as nanocarriers for DTX to enhance oral bioavailability and anti-tumor effects.

## 1. Introduction

Oral drug delivery is the oldest and most widely used route to administer medicines [1]. Compared with traditional systemic intravenous anti-tumor chemotherapy, the potential benefits behind developing novel oral anti-neoplastic drug delivery systems include [2] lower cancer therapy cost, tumor cell targetable treatment through gut-associated lymphoid tissue (GALT), enhancing the comfort and compliance of cancer patients or hospital-free treatment leading to “Chemotherapy at Home”, and eventually minimizing healthcare-associated severe infections or infection-related mortality [3]. However, in real-world cancer therapy, oral administration of many traditional chemotherapeutic agents has encountered significant challenges due to extensive pre-systemic metabolism [4], poor physicochemical properties of candidate drugs [5], high P-glycoprotein (P-gp) efflux transport [6], and low gastrointestinal cellular permeability [7].

With advancements in technologies pertaining to pharmaceutical sciences, various technological strategies, such as permeation enhancers, pro-drugs, and nanocarriers, have been employed to enhance the bioavailability of anti-neoplastic drugs after oral administration [7]. However, among the aforementioned approaches, nanocarrier-based oral targeting strategies, especially protein-based nanocarriers, have received tremendous attention, owing to their unique advantages, such as ease of biodegradability, extraordinary drug-binding capacity, and the presence of numerous functional groups available for chemical modifications [8,9].

Zein is a hydrophobic plant prolamin obtained from corn that exhibits a helical wheel-like structure consisting of nine homologous repeating units of polypeptides arranged in an anti-parallel manner [10]. Due to high levels of hydrophobic amino acids in the structure, zein is water-insoluble but soluble in more than 50% ethanol solutions. In recent years, zein nanoparticles encapsulating hydrophobic drugs have been demonstrated to increase the bioavailability and treatment effect of water-insoluble drugs, such as rapamycin [11], resveratrol [12], quercetin [13], and docetaxel [8,14]. Although zein NPs were applied to tumor-targeted drug delivery with a high drug-loading capacity [8] and controlled-release properties [14], satisfactory intestinal absorption and anti-tumor efficacy could not be attained as zein NPs displayed poor colloidal stability in biological fluids and limited selectivity to tumor cells [15]. However, recent studies have shown that numerous types of solute carrier transporters, such as glucose and L-amino acids, which are present throughout the gastrointestinal tract and tumor cell membranes [16], may represent potential targeting sites for successful oral delivery. 

Glucose (G), a small water-soluble molecule with multiple hydroxyl groups, is the primary energy source in the human body. It is speculated that G modification changes the surface properties of nanocarriers and their retention characteristics in blood. However, there have been no reports on the use of G to modify nanocarriers to enhance intestinal drug absorption.

Docetaxel (DTX), a water-insoluble and cytotoxic anti-neoplastic drug, has low oral bioavailability because of absorption hindrances, such as P-gp-mediated efflux and liver first-pass effect [17]. Several potential nanocarriers of DTX, including polymers [18], dendrimers [19], protein-based NPs, and liposomes [20], have been extensively investigated *in vitro* and *in vivo*. However, nanocarriers loaded with DTX based on natural proteins, such as zein, for oral administration have not been reported.

Hence, in the present study, a novel drug delivery system using hydrophilic G-modified zein (GZ) NPs for DTX (DTX-GNPs) was successfully designed and developed. We hypothesized that the introduction of targeting ligands, such as glucose to zein NPs, which can specially combine with relative transporters, may achieve a high tumor accumulation efficiency. *In vitro* and *in vivo* studies on cellular uptake and its mechanism, absorption site, pharmacokinetics, *ex vivo* distribution, and anti-tumor effects were also systematically conducted. 

## 2. Materials and Methods

### 2.1. Materials

DTX was obtained from Yi He Bioengineering Co., Ltd. (Beijing, China). Zein was purchased from Xiangtan Jiayeyuan Biotechnology Co., Ltd. (Hunan, China). Soybean lecithin was obtained from Taiwei Pharmaceutical Co., Ltd. (Shanghai, China). Coumarin-6, fetal bovine serum, penicillin, 0.25% trypsin-EDTA, streptomycin, non-essential amino acids, cell culture medium, and 3-(4,5-dimethylthiazol-2-yl)-2,5-diphenyltetrazolium bromide (MTT) were all purchased from Sigma (St. Louis, MO, USA). Dithiothreitol (DTT) was obtained from Bio Topped Co., Ltd. (Beijing, China). Ortho-phthalaldehyde (OPA), sodium dodecyl sulfate (SDS), and β-mercaptoethanol were purchased from boao tuoda technology Co., Ltd. (Beijing, China). All other chemicals and reagents used in the experiment were of analytical reagent grade or better.

CaCo-2 cell line and 4T1 cells were purchased from Ke Bai Biotechnology Co., Ltd. (Nanjing, China). BALB/c mice and Sprague Dawley rats (Female) [License No: scxk (Yu) 2017-0001] were obtained from the Laboratory Animal Center of Zhengzhou University (Zhengzhou, China).

### 2.2. Synthesis and Characterization of G-Functionalized Zein

#### 2.2.1. Synthesis of G-Functionalized Zein

As per a protocol from previous studies, G-functionalized zein was synthesized by the Maillard reaction [21]. In a typical process, 1.0 g of zein was dissolved in 85 mL of KCL-NaOH buffer solution (pH 13) and allowed to swell and alkalize under magnetic stirring for 1 h. After being totally dissolved, 2.0 g of G were added to the as-prepared zein solution and stirred for another 30 min. The mixture was then subjected to probe ultrasonication at 25 °C for 1 h. Thereafter, the obtained product mixture was purified *via* dialysis (molecular weight cut-off, 8000–12,000 Da) against ultrapure water at 25 °C for 24 h. Finally, G-functionalized zein was obtained following filtration and vacuum drying.

#### 2.2.2. Characterization of G-Functionalized Zein

Fourier transform infrared (FTIR) spectra of the synthesized GZ were recorded on a NICOLET 10 Fourier Transform Infrared spectrometer (Thermo Fisher Scientific, Waltham, MA, USA) using KBr pellets to compare the vibrational states in the scanning wave range of 4000–400 cm^−1^. The KBr pellet was composed of 5 mg of GZ and KBr mixed at a mass ratio of 1:100.

^1^H-NMR spectra of Z and GZ were measured using a superconducting nuclear magnetic spectrometer (BRUKER AVANCE-400M, Karlsruhe, Germany). Deuterated dimethyl sulfoxide (DMSO) and tetramethyl silane (TMS) were used as the solvent and internal standard, respectively.

The grafting degree of G in GZ was calculated by determining the number of free amino groups in GZ using the OPA method. Briefly, 80.0 mg OPA were dissolved in 2 mL of methanol, and then 5.0 mL 20% SDS, 50.0 mL borax (0.1 mol/L), and 200 μL β-mercaptoethanol were added to the mixture. The mixture was then diluted to 100 mL with distilled water and then to an 8.0 mL mixture solution and 0.4 mL solution before or after Z reacting with G were mixed; the reaction was carried out in a water bath (35 °C) for 2 min. Thereafter, the absorbance of the reaction solution was measured at a 340 nm wavelength. A standard curve was plotted with 0.1 to 0.5 mg/mL lysine solution to calculate the free amino group content in the sample. The degree of grafting (DG) was calculated using the following equation: DG = (C_0_ − C_t_)/C_0_ × 100. C_0_ represents the content of free amino groups in the Z solution before adding G during the synthesis of GZ; C_t_ represents the content of free amino groups in the solution after Z reacts with G during the synthesis of GZ (as described in Section 2.2.1).

The isoelectric point was analyzed using the precipitation method. Z or GZ (100 mg) was dissolved in 40 mL of KCl-NaOH buffer solution (pH 13), and the obtained Z or GZ solution was centrifugated at 10,000 rpm for 10 min to remove any insoluble ingredients. Z or GZ solution (1 mL) was titrated with 1 mol/L hydrochloric acid (HCl) solution until precipitation occurred. After centrifugation at 10,000 rpm for 10 min, the pH of the supernatant was measured using a pH meter, and the sediment was dried and weighed. The pH corresponding to the highest precipitation was regarded as the isoelectric point.

### 2.3. Preparation and Characterization of DTX-GNPs

#### 2.3.1. Preparation of DTX-GNPs

DTX-GNPs were fabricated using the solvent evaporation method, as previously described [20]. Z, GZ, and PC-80 (5:1:6, *w*/*w*) were dissolved in 10 mL 70% ethanol under magnetic stirring for 1 h. Next, 1 mL of DTX (15 mg/mL) or coumarin 6 (C6) solution was added to the ethanol solution. Thereafter, the prepared solution containing DTX or coumarin 6 was slowly added dropwise to 25 mL of ultrapure water within 10 min under magnetic stirring. The flask was then placed in a water bath and allowed to react for 1.5 h to remove ethanol. Finally, the resulting NP suspensions were filtered using a syringe filter (0.45 μm) to eliminate unreacted materials, yielding DTX-GNPs or DTX-NPs. DTX-GNPs corresponding to different feed ratios of GZ and Z (1:0, 1:3, 1:5, and 1:6) were labeled as DTX-GNPs (1:0), DTX-GNPs (1:3), DTX-GNPs (1:5), and DTX-GNPs (1:6), respectively.

#### 2.3.2. Characterization of DTX-GNPs

The average particle size, distribution, and zeta potential values of the different DTX nanoparticles were assessed using a laser nanoparticle size analyzer (Nano-ZS 90, Malvern, UK) at room temperature (25 °C). Zeta potential was measured as previously described [20]. The surface morphology of DTX-GNPs was observed using a transmission electron microscope (TEM; JEM-100CXfl, Tokyo, Japan). The DTX-GNP suspension was fixed on a copper grid covered with formvar film, stained with phosphotungstic acid, and observed under a microscope at an accelerating voltage of 120 kV. The encapsulation efficiency (EE) and drug loading (DL) of DTX-GNPs were measured as previously mentioned [22]. DTX concentration was detected using a high-performance liquid chromatography (HPLC) system equipped with an ultraviolet detector and a symmetry-C18 column (4.6 mm × 250 mm, 5 μm). The chromatographic conditions for HPLC were set as follows: detection wavelength, 229 nm; flow rate, 1.0 mL/min; flow phase, water: methanol (25:75, *v/v*); injection volume, 20 μL; column temperature, 30 °C. The EE and DL of DTX were calculated using the following equations: EE (%) = [(W_total_ − W_free_)/W_total_] × 100 % and DL = [(W_total_ − W_free_)/ (W_total—_W_free_ + W_carrier_)] × 100%, where W_carrier_ is the weight of carriers added to the system.

#### 2.3.3. *In Vitro* Release Behavior of DTX-GNPs

DTX release patterns of different DTX preparations were assessed using the dialysis method. First, DTX preparations were sealed in dialysis bags (molecular weight cut-off, 8000–14,000 Da) and incubated on a shaker (ZD-85, Zhejiang, China) with moderate shaking (100 rpm) at 37 °C. HCl solution (pH 1.2) with pepsin was used as simulated gastric fluid (SGF) and phosphate-buffered solution (PBS) (pH 6.8) with trypsin was used as simulated intestinal fluid (SIF). It is worth noting that the entire volume of the release medium SGF was displaced by SIF after 2 h. Tween 80 (0.5%), as a solubilizer, was added to SGF and SIF to meet sink conditions. At pre-determined time points, 1 mL of release medium was withdrawn and filled with fresh media each time. The amount of DTX released into the release medium was determined *via* HPLC, as described in Section 2.3.2, to obtain cumulative release profiles.

### 2.4. Cellular Uptake and the Underlying Mechanism

Cellular accumulation of DTX preparations was qualitatively and quantitatively evaluated using fluorescently labeled nanoparticles in CaCo-2 cells. CaCo-2 cells were seeded in 24-well plates at a density of 1 × 10^5^ cells/well and incubated for 24 h at 37 °C. The growth medium was then replaced with a growth medium containing coumarin 6 (C6)-labeled DTX nanoparticles. After incubation for 1, 2, 3, and 4 h at 37 °C, the cells were washed with PBS three times. For qualitative analysis, intracellular fluorescence intensity was observed using an inverted fluorescence microscope (BH-2, Olympus Corporation, Tokyo, Japan). For quantitative analysis, the cells were treated with 500 μL of cell lysis solution and incubated for another 2 h at 37 °C. Thereafter, the lysate (100 μL) was mixed with DMSO to measure the C6 content at 490 nm using a SpectraMax M5 microplate reader (Molecular Devices LLC, San Jose, CA, USA), and 100 μL of lysate was used to determine the protein content using a BCA protein assay kit, according to the manufacturer’s protocol.

The cellular uptake mechanism of DTX nanoparticles was the same as the quantitative assay method described above, except for the three competitive inhibitors (Chlorpromazine, 0.3 mg/mL; Amiloride, 12 μg/mL; Indomethacin, 36 μg/mL) or absorption inhibitors (D-glucose solution, 5 mg/mL) were co-incubated with the cells for 30 min before the experiment [23].

### 2.5. Intestinal Absorption Site 

The absorption sites of DTX-GNPs and DTX-NPs in the small intestine of rats were investigated using in situ rat intestinal circulation experiments [24]. After fasting for 12 h, the rats were anesthetized and placed in a supine position under an infrared lamp to maintain a normal temperature. A midline longitudinal incision was carefully made in the abdomen, and intestinal segments approximately 8–12 cm in the duodenum, jejunum, or ileum were cannulated with perfusion tubing. The intestinal segments were gently washed with a pre-heated saline solution and equilibrated with KR buffer solution. Thereafter, the circulation experiment was performed at a flow rate of 2 mL/min for 4 h. A KR buffer solution containing DTX-NPs and phenol red (0.02 mg/mL) was used as the circulating solution. At the planned time points, 1 mL of the circulating sample was replaced with fresh pre-warmed solution. The concentration of phenol red was detected after sample filtering through a 0.45-μm filter membrane. To determine the concentrations of phenol red, dual-wavelength spectrophotometry was employed to correct the water volume at the detection wavelength and reference wavelengths of 558 nm and 598 nm, respectively. The amount of DTX in the sample was quantified using HPLC, as described in Section 2.3.2. 

The absorption rate constant (*k_a_*) was calculated based on the concentration of DTX in the circulating solution, according to the slope of the straight line obtained by linear regression of lg(C_0_V_0_ − C_t_V_t_) to time, where C_0_ and C_t_ are the DTX concentrations in the perfusate corresponding to the initial time and time t, respectively. V_0_ and V_t_ represent the perfusate volumes corresponding to the initial time and time t, respectively.

### 2.6. Pharmacokinetic Studies

Pharmacokinetic studies of different DTX preparations were performed in healthy Sprague Dawley rats (weight, 200 ± 20 g; female; SCXK(YU)2017-0001, No. DW2020060051). Thirty-six Sprague Dawley rats were randomly assigned to six groups (*n* = 6) and fasted for 12 h before the experiment: (1) intravenous (i.v.) Duopafei^®^ injection (10 mg/kg), (2) p.o. DTX-NPs, (3) p.o. DTX-GNPs (1:0), (4) p.o. DTX-GNPs (1:3), (5) p.o. DTX-GNPs (1:5), and (6) p.o. DTX-GNPs (1:6). The dose in each experimental group was 20 mg/kg. At pre-determined time intervals, orbital blood (0.5 mL) was collected and centrifuged at 3000 rpm for 5 min. The supernatant (0.2 mL) was placed in a 5 mL centrifuge tube, and 1.5 mL methyl tertbutyl ether were added, followed by vortexing and centrifugation. The supernatant (2.0 mL) was dried under air flow at 40 °C. The dry residue was re-dissolved in methanol (0.2 mL) and subjected to HPLC, as described in Section 2.3.2. The pharmacokinetic analysis was performed using a non-compartmental model within the PK Solver software (PKSlover 3.0, China Pharmaceutical University, Nanjing, China).

### 2.7. Biodistribution and Anti-Tumor Effect

Tumor-bearing mice, as orthotopic tumor models, were fabricated by subcutaneous injection of 4T1 cell suspension (200 μL, 5 × 10^6^ cells) into the right shoulder of BALB/c female mice (18–20 g). After 10 days, the mice were used for experiments when the tumor volume reached 80–100 mm^3^.

For the biodistribution study, 24 tumor-bearing mice were randomly assigned to three groups and treated with IR-780 iodide-labeled DTX-NPs according to a method described previously [25]. IR-780 solution (composed of polyoxymethylene castor oil, ethanol, and 5% glucose solution at a mass ratio of 1:1:9) was assigned as the control group; IR780-DTX-NPs and IR780-DTX-GNPs were assigned as study groups (the preparation method was the same as that of C6 coated nanoparticles). After fasting overnight with free access to water, the mice were administered the aforementioned formulations. At pre-determined time points, the mice were sacrificed, and tissues, such as those from the heart, liver, spleen, lung, kidneys, stomach, intestine, and tumor, were excised and thoroughly washed with normal saline. IR-780 iodide imaging experiments were performed at 1, 2, 6, 12, and 24 h post-administration using an *in vivo* imaging system (Quick View 3000 Bio-Real, Salzburg, Austria) equipped with an excitation bandpass filter at 720 nm and an emission filter at 790 nm. Fluorescence intensity was measured using a SpectraMax M5 microplate reader at 780 nm [20].

To determine the anti-tumor effects, 32 tumor-bearing mice were randomly divided into four groups (*n* = 8) and treated with the following formulations: (1) p.o. saline solution (200 μL), (2) i.v. Duopafei^®^ (10 mg/kg), (3) p.o. DTX-NPs (20 mg/kg), and (4) p.o. DTX-GNPs (1:5) (20 mg/kg). The mice were treated once daily, and tumor size and body weight were measured every second day. After 14 days of treatment, the mice were euthanized by cervical dislocation after collecting blood (0.5 mL) from the lateral canthus, and the tumor mass was harvested, photographed, and weighed. Within 0.5 h after blood collection, the white blood cell count was measured [26]. The tumor volumes and growth inhibition rates, which were used to assess anti-tumor efficacy, were calculated according to the following equation: tumor volume = (tumor length) × (tumor width)^2^/2; tumor inhibition rate = ((Ws − Wt)/Ws) × 100%), where Ws and Wt are the average tumor weights in the saline solution group and other groups, respectively.

### 2.8. Statistical Analysis

All data are expressed as the mean ± standard deviation (S.D.) unless otherwise noted. Statistical analysis was performed using the Student’s *t*-test or one-way analysis of variance (ANOVA). Differences were considered statistically significant when *p*-values were less than 0.05.

## 3. Results 

### 3.1. Synthesis and Characterization of G-Functionalized Zein

G-functionalized zein was successfully synthesized in one step by the Maillard reaction, and its chemical structure was confirmed using FTIR and NMR spectra (Figure 1A,B). Compared with Z, GZ showed two new strong absorption peaks at 1260 and 1023 cm^−1^. In other words, an absorption peak at 1260 cm^−1^ was produced by the stretching vibration of the C–N bond formed between G and Z, whereas another absorption peak at 1023 cm^−1^ was produced by the stretching vibration of the C–O bond in G. Furthermore, there were two characteristic absorption peaks of the amino group in Z at 1646 and 1539 cm^−1^. However, the two characteristic absorption peaks of the amino group in GZ changed. One peak shifted to 1632 cm^−1^, and the other peak weakened. These changes indicate that the amino groups in Z reacted with the carbonyl group in G. These results confirmed that G successfully bound to Z. According to the NMR spectra of Z and GZ (Figure 1B), the H content in Z increased after G modification. Compared with the ^1^H-NMR spectrum of Z, the signal of GZ at 4.39 ppm indicated the proton of the –OH group in G, and the signal at 2.97 ppm indicated the proton of the –CO–CH_2_– group in GZ. In addition, the signal at 3.63 ppm corresponded to the proton of the –CH_2_–O– group in G. Thus, GZ was successfully synthesized. 

The grafting degree of G in Z was 31.53%. The solubility of Z or GZ was the lowest at the isoelectric point, which was determined as shown in Figure 1C. The isoelectric point of Z and GZ was 6.8 and 7.54, respectively.

### 3.2. Characterization of DTX-GNPs

The characteristics of the different DTX-NPs are summarized in Table 1. G modification had a slight effect on particle size and polydispersity index (PDI), but it had no significant effect on zeta potential, drug loading (DL), and encapsulation efficiency. The particle size of these NPs was less than 140 nm, with a PDI of <0.3, a negative zeta potential of approximately −20.0 mV, and a DL and encapsulation efficiency of approximately 5.90 % and 85.0%, respectively. Glucose-modified density directly affects the oral bioavailability of nanoparticles. When the DTX-GNPs were prepared with GZ and Z according to the quality ratio of 1:5, the epithelial cells displayed the best uptake effect, with the highest oral bioavailability; therefore, this ratio was used to study the biological distribution and pharmacodynamics of nanoparticles *in vivo.* For DTX-GNPs (1:5), the size and zeta distribution are shown in Figure 2A,B, respectively. Transmission emission microscopy revealed that the DTX-GNPs were nearly spherical with a particle size of approximately 100 nm (Figure 2C). 

DTX released from the different preparations is shown in Figure 2D. To simulate the physiological characteristics of gastrointestinal pH, the release characteristics of these NPs were studied in an HCl solution (pH 1.2) for the first 2 h and then in a PBS solution (pH 6.8) for the next 70 h. As shown in Figure 2D, both DTX-GNPs (1:5) and DTX-NPs showed sustained release characteristics, particularly the former. 

### 3.3. Cell Experiments

#### 3.3.1. Cellular Uptake

C6-labeled DTX-GNPs were used for visualizing cellular uptake. Uptake results of different preparations in CaCo-2 cells are shown in Figure 3A. Over time, the fluorescence intensity of the cells increased as the transport of C6-labeled NPs increased. Figure 3A shows that DTX-GNPs (1:5) exhibited a stronger fluorescence intensity than DTX-NPs at all time points, suggesting that G modification significantly enhanced the cellular uptake of DTX-GNPs. Further quantitative analysis revealed that DTX-GNPs (1:5) were the highest compared with DTX-GNPs (1:6) and DTX-GNPs (1:3), implying that the modification ratio of G may also be an important factor affecting the cellular uptake of DTX-GNPs (Figure 3B).

#### 3.3.2. Cellular Uptake Mechanism

To further elucidate the cellular uptake mechanism of DTX-GNPs (1:5) by CaCo-2 cells, D-glucose and three endocytic inhibitors (chlorpromazine, amiloride, and indomethacin) were employed to investigate the inhibitory effect of G modification. As shown in Figure 3C, compared with the control group, the pre-incubation with D-glucose solution hardly affects the uptake of DTX-NPs by CaCo-2 cells but significantly reduces the uptake of DTX-GNPs, especially in the DTX-GNPs (1:5) group where inhibition rates of 17.00% were attained. The possible mechanism is that the D-glucose binds to the glucose transporter in the pre-incubation phase, resulting in a significant reduction in uptake of DTX-GNPs, suggesting that glucose transporters can mediate the cell uptake of glucose-modified nanoparticles. In addition, when compared to the control group, the addition of chlorpromazine reduced the uptake of DTX-NPs, DTX-GNPs (1:6), DTX-GNPs (1:5), and DTX-GNPs (1:3) by 8.49%, 17.86%, 21.38%, and 13.57%, respectively; the addition of amiloride reduced the uptake of the above nanoparticles by 15.66%, 19.34%, 22.18%, and 19.63%, respectively; the addition of indomethacin reduced the uptake of the above nanoparticles by 24.98%, 24.08%, 32.56%, and 28.33%, respectively. These results indicated that the caveolin pathway plays an important role in mediating the endocytosis mechanism of nanoparticles. The inhibition of the endocytosis difference of the four nanoparticles may be due to the modification of glucose affecting the endocytosis pathway of the nanoparticles. 

### 3.4. Intestinal Absorption Site

The absorption rate constant (*K_a_*) of DTX in different DTX-NP preparations in each intestinal segment is shown in Figure 3D. The *K_a_* value of DTX-NPs in the duodenum was significantly higher than that of DTX-GNPs (1:5) (Figure 3D). However, the *K_a_* values of DTX-GNPs (1:5) in the jejunum and ileum were 1.07- and 4.38-fold higher than those of DTX-NPs, respectively. These results indicate that G modification altered the absorption sites of DTX-NPs and significantly enhanced their absorption in the jejunum and ileum.

### 3.5. Pharmacokinetic Studies

The corresponding pharmacokinetic (PK) parameters are listed in Table 2. From Table 2, the oral bioavailability of the DTX-NPs group and DTX-GNPs (1:5) group was 43.82% and 96.04%, respectively. Compared with the DTX-NPs group, the modification of glucose significantly enhanced the oral absorption and the bioavailability of DTX-NPs, and the absolute bioavailability of DTX-GNPs (1:5) was 2.19 times higher than DTX-NPs. Similarly, the T_max_ of the DTX-GNPs (1:5) was extended to 2 h, the MRT increased by 8.07 h, and the AUC_0–24h_ was 2.19 times greater than DTX-NPs. In addition, among the different preparation groups, the oral bioavailability was also varied from the grafting degree of G. Among them, the absolute oral bioavailability of DTX-GNPs (1:5) was 2.86, 1.54, and 1.95 times higher than DTX-GNPs (1:0), DTX-GNPs (1:3), and DTX-GNPs (1:6), respectively, indicated that the glucose density on the surface of the nanoparticles directly affected the oral bioavailability, and showed the trend of first rising and then decreasing with the increase of glucose density. 

The pharmacokinetic behavior of different DTX preparations after oral administration was evaluated in rats (Figure 3E). As shown in Figure 3E, DTX-NPs exhibited a shorter peak time (Tmax) and higher peak concentration (Cmax) than other preparations. However, the plasma concentration of DTX-GNPs (1:5) from 0 h to 24 h was significantly higher than that of DTX-NPs. Compared with DTX-NPs, DTX-GNPs (1:5) prolonged the mean residence time (MRT) from 20.91 to 25.15 h and improved the oral bioavailability from 43.82% to 96.04%. These results indicate that G modification significantly delayed the peak time (Tmax) and prolonged the MRT, thereby improving the oral bioavailability of DTX-NPs. According to the test results, the feed ratio of GZ and Z affected oral bioavailability. The maximum bioavailability was achieved at a feed ratio of 1:5. The DTX-GNPs with a feed ratio of 1:0 resulted in the lowest oral bioavailability. 

### 3.6. Biodistribution and Anti-Tumor Effect

The *ex vivo* images of the tissues and organs of tumor-bearing mice after oral administration of different DTX preparations labeled with IR-780 iodide are shown in Figure 4A. DTX-GNPs (1:5) and DTX-NPs showed strong fluorescence for at least 12 h in the small intestine, indicating that Z and GZ prolonged the retention of DTX-GNPs (1:5) and DTX-NPs in the small intestine (Figure 4A). Stronger fluorescence signals were observed at the tumor site for DTX-GNPs (1:5), compared to those for DTX-NPs, at most time points. However, weaker fluorescence signals were observed in the liver, kidneys, and lungs for DTX-GNPs (1:5) compared to those for DTX-NPs. These qualitative results were further confirmed by quantitative analysis (Figure 4B and Table 3). The results indicated that G modification enhanced the tumor targeting of DTX-NPs and reduced toxicity in the liver, kidneys, and lungs. 

The anti-tumor activities of DTX-NPs and DTX-GNPs (1:5) were also studied in a subcutaneous 4T1 model (Figure 5A,B). The tumor growth inhibition rate was 89.81% in the DTX-GNP (1:5) group and 77.34% in the DTX-NP group. During the experiments, the body weights of mice in the saline group and the Duopafei group were substantially lower than those of mice in other groups. Although the tumor volume decreased, the body weights of the mice remained unchanged in the DTX-NP and DTX-GNP (1:5) groups, and the white blood cell count was normal. These results confirmed that G modification enhanced the anti-tumor effect of DTX-NPs without any obvious toxicity. Statistical test method used in this figure, One way ANOVA.

## 4. Discussion

To the best of our knowledge, there have been no studies on the regulation of oral absorption of G-modified drug-loaded nanoparticles by GLUT. In this study, we successfully developed a novel drug delivery system for DTX-GNPs (1:5). DTX-GNPs (1:5) not only significantly increased the oral bioavailability of DTX but also enhanced the anti-tumor efficacy due to G modification.

G modification altered the isoelectric point of hydrophobic zein from 6.80 to 7.54, thereby altering the surface properties and intestinal absorption sites of DTX-NPs. The residence time of DTX-NPs in the duodenum is a period of only tens of seconds after oral administration, which is very short and not conducive to the adequate absorption of DTX-NPs in the duodenum. Compared to DTX-NPs, DTX-GNPs (1:5) showed improved absorption in the entire small intestine. Therefore, it was inferred that DTX-NPs was absorbed mainly in the duodenum and that DTX-GNPs (1:5) would be better absorbed in the entire intestine compared to DTX-NPs; these findings were confirmed by the results of pharmacokinetic analyses. After oral administration, DTX-NPs showed a shorter peak time and higher peak concentration than DTX-GNPs (1:5). G modification prolonged the absorption time of DTX-NPs, which may be attributed to the better absorption of DTX-GNPs (1:5) in the entire small intestine. The plasma concentration of DTX-GNPs (1:5) was significantly higher than that of DTX-NPs alone. Correspondingly, the oral bioavailability of DTX-NPs was significantly enhanced by G modification. These results were closely related to the different absorption sites of the two nanoparticles. The change in their absorption sites is related to the distribution and mediation of G transporters in the small intestine.

G modification significantly enhanced CaCo-2 cell uptake of DTX-NPs and oral bioavailability *via* G transporters, and uptake was significantly affected by the density of G on the surface of DTX-NPs or the mass ratio of GZ:Z from 1:6 to 1:0. The CaCo-2 cell uptake and oral bioavailability of DTX-NPs reached their maximal levels at a mass ratio of 1:5. G transporters in intestinal epithelial cells are usually limited. With an increase in the mass ratio of GZ:Z, cellular uptake *via* the binding of G and its transporters gradually saturated, and oral bioavailability peaked. When the mass ratio of GZ:Z continued to increase, steric hindrance possibly interfered with the recognition of nanoparticles in a manner similar to the results reported by Jain et al. [27] and Li et al. [28]. Galactosylation NPs can enhance cellular uptake and promote the oral absorption of some drugs. G modification not only enables nanoparticles to target G transporters but also enhances endocytosis by CaCo-2 cells. G is a water-soluble polyhydroxy compound, and Z is a hydrophobic compound. G modification altered the surface properties of DTX-GNPs, such as hydrophilicity, thus enhancing endocytosis *via* other pathways. The intestinal absorption of ligand-modified nanocarriers is not dependent on a single mechanism, and multiple absorption mechanisms often coexist.

Compared with DTX-NPs, DTX-GNPs (1:5) showed improved anti-tumor efficacy. During the administration period, the activity of the tumor-bearing mice was normal, mice appeared in good condition, and their body weights were constant due to enhanced absorption in the intestine and bioavailability, reduced adsorption of plasma proteins, and enhanced tumor targeting *via* G modification. According to previous studies [29], intact nanoparticles (small amounts) can be transported from the intestinal tract into the blood. G modification of DTX-NPs enhanced their surface hydrophilicity, reduced the adsorption of plasma proteins, and prevented phagocytosis by the monocyte–macrophage system. They were primarily distributed in tumors, owing to their enhanced permeability and retention (EPR) effect, which could contribute to the enhanced anti-tumor efficacy of DTX-GNPs (1:5). Furthermore, several G transporters are present in tumor cells. With the help of G transporters, DTX-GNPs (1:5) were more likely to be absorbed by tumor cells than DTX-NPs, further enhancing their anti-tumor effect [30]. DTX-GNPs showed not only a better anti-tumor effect but also no obvious systemic toxicity, which was related to tumor passive and active targeting by G modification.

## 5. Conclusions

In the present study, a novel delivery system, G-modified zein nanoparticles for DTX, was successfully prepared and characterized *in vitro* and *in vivo*. Compared with DTX-NPs, G-modification significantly enhanced Caco-2 cellular uptake of the drug *via* G transporters and other endocytic pathways, intestinal absorption in the jejunum and ileum, and oral bioavailability. G-modification decreased drug distribution in some organs, such as the liver, lungs, and kidneys, and enhanced tumor-targeting and anti-tumor effects without obvious toxicities. The use of G-modified zein nanoparticles may be a promising strategy to enhance the oral bioavailability and antitumor effects of DTX.

## Figures and Tables

**Figure 1 pharmaceutics-14-01361-f001:**
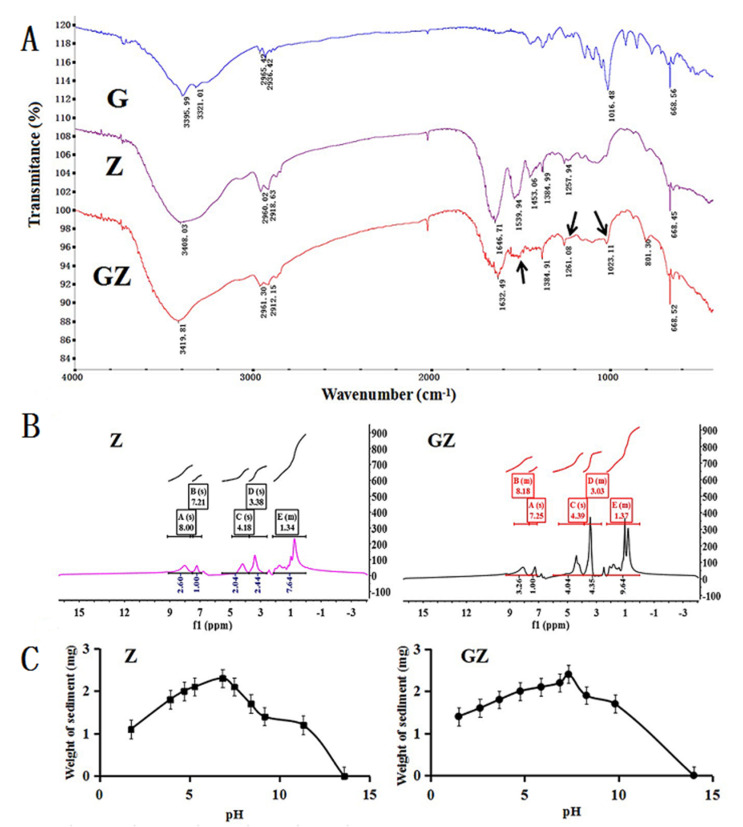
The characterization of GZ synthesized. (**A**) The FTIR spectroscopy curve. (**B**) The ^1^H−NMR spectrum of Z and GZ. (**C**) The isoelectric point of Z and GZ (mean ± S.D., *n* = 3).

**Figure 2 pharmaceutics-14-01361-f002:**
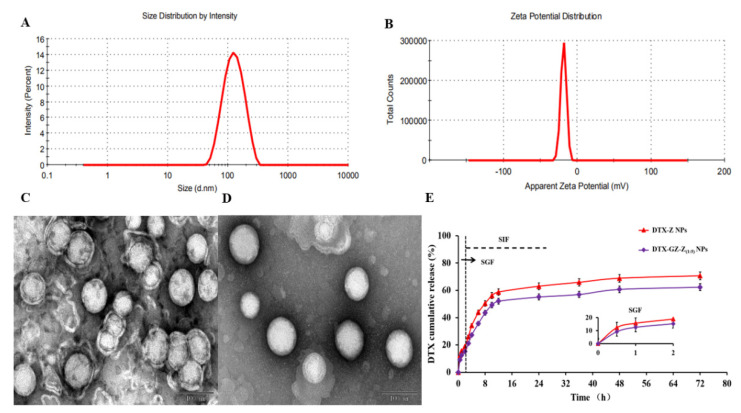
*In vitro* characterization of DTX-GNPs (1:5). (**A**) The distribution of DLS particle size. (**B**) Zeta potential distribution. (**C**) The TEM image of DTX-NPs (Scale bar represents 100 nm). (**D**) The TEM image of DTX-GNPs (Scale bar represents 100 nm). (**E**) *In vitro* release in pH 1.2 HCl solution for the first 2 h and then in pH 6.8 PBS solution (mean ± S.D., *n* = 3).

**Figure 3 pharmaceutics-14-01361-f003:**
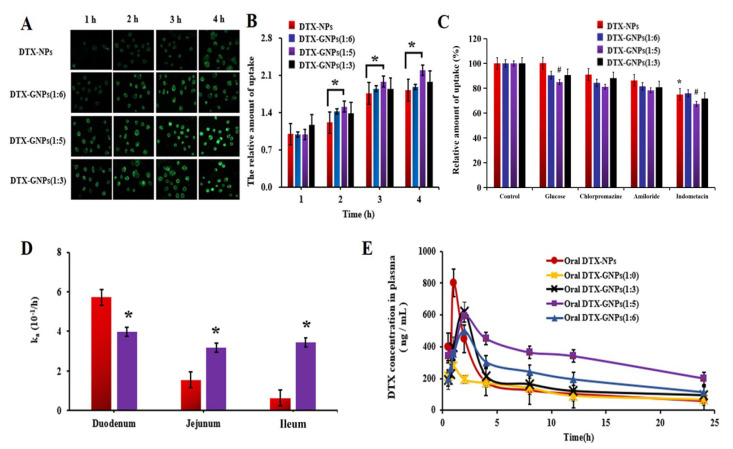
The CaCo-2 cells uptake and mechanism study of different DTX nanoparticles labeled with C6. (**A**) The intracellular fluorescence imaging at different time points (200×) (the scale bar represents 10 μm). (**B**) Relative cell uptake amount after incubation for different times (mean ± S.D., *n* = 3); (**C**) Inhibition of G and endocytosis inhibitors on the cell uptake after incubation for 3 h (mean ± S.D., *n* = 3). * *p* < 0.05 compared with DTX-NPs, # *p* < 0.05 compared with DTX-GNPs (1:5); (**D**) Absorption rate constants (ka) of different DTX nano-particles in different intestinal segments of rats (* *p* < 0.05 compared with oral DTX-NPs); (**E**) Plasma concentration-time profiles in rats after intragastric administration of different DTX nanoparticles (20 mg/kg) (mean ± S.D., *n* = 6). Statistical test method used in this figure, One way ANOVA and post hoc test for multiple comparisons.

**Figure 4 pharmaceutics-14-01361-f004:**
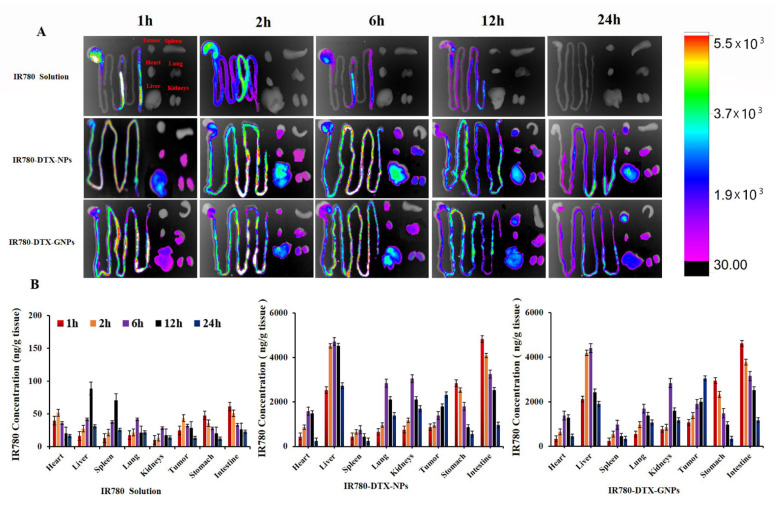
*Ex vivo* distribution of different nanoparticles labeled with IR-780 iodide after intragastric administration in tumor-bearing mice. (**A**) Fluorescent imaging. (**B**) Quantitative analysis based on fluorescence intensity (mean ± S.D., *n* = 3).

**Figure 5 pharmaceutics-14-01361-f005:**
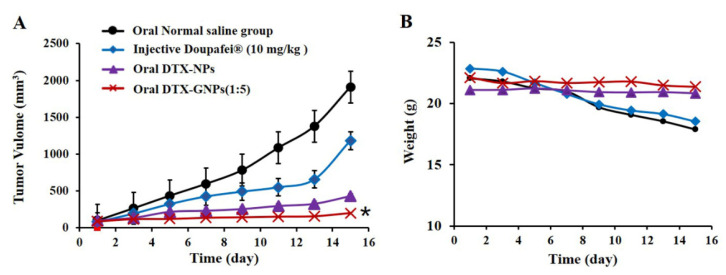
*In vivo* antitumor activity of different DTX preparations against mouse 4T1 breast tumor xenograft. (**A**) Growth curves of tumor in the mice (* *p* < 0.05 compared with oral DTX-NPs). (**B**) Body weight of the mice in each group (mean ± S.D., *n* = 8). Statistical test used in this figure, One way ANOVA.

**Table 1 pharmaceutics-14-01361-t001:** The characterization of different DTX preparations (mean ± S.D., *n* = 3) *.

Various Nanoparticles	Particle Size (nm)	PDI	Zeta Potential (mV)	DL (%)	EE (%)
DTX-NPs	108.1 ± 5.3	0.133 ± 0.012	−22.0 ± 2.72	5.45 ± 0.17	78.23 ± 2.45
DTX-GNPs	87.9 ± 3.2	0.126 ± 0.008	−18.5 ± 2.15	6.05 ± 0.19	86.73 ± 2.67
DTX-GNPs (1:6)	131.3 ± 4.8	0.206 ± 0.023	−20.8 ± 3.46	5.56 ± 0.09	79.65 ± 1.35
DTX-GNPs (1:5)	123.7 ± 5.1	0.287 ± 0.011	−20.0 ± 2.07	5.89 ± 0.14	84.39 ± 2.04
DTX-GNPs (1:3)	136.4 ± 6.5	0.078 ± 0.037	−19.1 ± 1.25	5.95 ± 0.22	85.23 ± 3.15

* Statistical test method used in this table, One way ANOVA.

**Table 2 pharmaceutics-14-01361-t002:** The Pharmacokinetic Parameters (PKP) in rats after administration of different DTX preparations (mean ± S.D., *n* = 6) ^#^.

Pkp	Duopafei^®^ (Injection, 10 mg/kg)	Oral (20 mg/kg)				
		DTX-Nps	DTX-Gnps (1:0)	DTX-Gnps (1:6)	DTX-Gnps (1:5)	DTX-Gnps (1:3)
C_max_ (ng/mL)	7106.35 ± 253.78	806.63 ± 78.43	288.45 ± 29.32	497.56 ± 51.67	596.67 ± 74.81	620.28 ± 93.15
T_max_ (h)	0.02 ± 0.01	1.00 ± 0.21	1.00 ± 0.24	2.00 ± 0.12	2.00 ± 0.16	2.00 ± 0.20
AUC_0–24h_ (ng/mL·h)	4205.48 ± 391.87	3685.71 ± 414.76	2825.38 ± 328.87	5253.33 ± 621.72	8077.53 ± 743.16 *	4146.21 ± 511.34
MRT (h)	0.33 ± 0.03	17.45 ± 2.85	20.91 ± 2.34	20.99 ± 2.65	25.15 ± 2.21 *	23.26 ± 2.51
F (%)	100.00	43.82	33.59	62.46	96.04 *	49.30

* Significant differences compared to DTX-NPs (*p* < 0.05). ^#^ Statistical test method used in this table, One way ANOVA and post hoc test for multiple comparisons.

**Table 3 pharmaceutics-14-01361-t003:** The AUC_0–24h_ of IR 780 in the organs and tissues of 4T1 tumor-bearing mice after oral administration of IR 780 solution, IR 780-NPs, and IR 780-GNPs (1:5) (mean ± S.D., *n* = 8) ^.

The Organs/Tissues	AUC_0–24h_ (μg/mL·h)		
	IR780 Solution	IR780-Nps	IR780-Gnps (1:5)
Heart	0.62 ± 0.13	5.09 ± 0.05 ^#^	4.63 ± 0.06 ^#^*
Liver	1.09 ± 0.25	18.91 ± 0.84 ^#^	13.60 ± 0.35 ^#^*
Spleen	0.92 ± 0.05	2.28 ± 0.01 ^#^	2.51 ± 0.03 ^#^
Lung	0.61 ± 0.19	8.94 ± 0.22 ^#^	6.08 ± 0.16 ^#^*
Kidneys	0.43 ± 0.30	9.65 ± 0.25 ^#^	7.72 ± 0.23 ^#^*
Stomach	0.53 ± 0.02	5.86 ± 0.07 ^#^	5.40 ± 0.04 ^#^
Intestine	0.73 ± 0.36	12.01 ± 0.32 ^#^	11.96 ± 0.39 ^#^
Tumor	0.63 ± 0.15	8.09 ± 0.12 ^#^	10.11 ± 0.17 ^#^*

^#^ Significant differences compared with IR780 solution (*p* < 0.05). * Significant differences compared with IR780-NPs (*p* < 0.05). ^ Statistical test used in this table, One way ANOVA and post hoc test for multiple comparison.

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
