# Peer review of "Glucose-Modified Zein Nanoparticles Enhance Oral Delivery of Docetaxel"

_pharmaceutics, 2022, doi:10.3390/pharmaceutics14071361_

Round 1

Author Response

Responds to the reviewer’s comments and Suggestions:

Reviewer 1:
The manuscript “Glucose-modified zein nanoparticles enhanced effectively oral delivery of docetaxel” describes the development, characterization, in vitro and in vivo pharmacological profile of glucose-zein nanoparticles loaded with docetaxel for oral delivery. The authors performed several experiments but the English must be revised in all the manuscript. Moreover, the following criticisms should be addressed:
Answer:
Thank you for your valuable advice. According to your comments and suggestions, we have made every effort to improve the quality and clarity of the language throughout the manuscript and the manuscript have been polished by a professional assistant in writing before re-submitted to Pharmaceutics.  

Editing Certificate, please see the attachment.

Question 1:
What is the advantage of using this type of oral formulation instead of the systemic route commonly used for antineoplastic drugs? It would be better to clarify this concept in the introductory section and compare it with other published data regarding the use of Docetaxel-containing zein nanoparticles.
Answer:
Thank you for your instructive suggestions. We have added the relative description in the introductory section and the following description was excerpted from the manuscript.
Oral drug delivery is the oldest and most widely used route to administer medicines. Compared with the traditional systemic intravenous anti-tumor chemotherapy, the potential benefits behind developing novel oral antineoplastic drug delivery systems include lower cancer therapy cost, tumor cell targetable treatment through gut-associated lymphoid tissue (GALT), enhancing the comfort and compliance of cancer patients or hospital-free treatment leading to “Chemotherapy at Home”, and eventually minimizing healthcare-associated severe infections or infection-related mortality. However, in real-world cancer therapy, oral administration of many traditional chemotherapeutic agents has encountered significant challenges due to extensive pre-systemic metabolism, poor physicochemical properties of candidate drugs, high P-glycoprotein (P-gp) efflux transport, and low gastrointestinal cellular permeability.
With advancements in technologies pertaining to pharmaceutical sciences, various technological strategies, such as permeation enhancers, pro-drugs, and nanocarriers, have been employed to enhance the bioavailability of anti-neoplastic drugs after oral administration. However, among the approaches, nanocarrier-based oral targeting strategies, especially protein-based nanocarriers, have received tremendous attention, owing to their unique advantages, such as ease of biodegradability, extraordinary drug binding capacity, and the presence of numerous functional groups available for chemical modifications.
Zein is a hydrophobic plant prolamin obtained from corn that exhibits a helical wheel-like structure consisting of nine homologous repeating units of polypeptides arranged in an anti-parallel manner. Due to high levels of hydrophobic amino acids in the structure, zein is water-insoluble but soluble in more than 50% ethanol solutions. In recent years, zein nanoparticles encapsulating hydrophobic drugs have been demonstrated to increase the bioavailability and treatment effect of water-insoluble drugs, such as rapamycin, resveratrol, quercetin, and docetaxel. Although zein NPs were applied to tumor-targeted drug delivery with high drug-loading capacity and controlled-release properties, satisfactory intestinal absorption and anti-tumor efficacy could not be attained, as zein NPs displayed poor colloidal stability in biological fluids and limited selectivity to tumor cells. However, recent studies have shown that numerous types of solute carrier transporters, such as glucose and L-amino acids, which are present throughout the gastrointestinal tract and tumor cell membranes, may represent potential targeting sites for successful oral delivery.
Question 2:   
Section 2.3.1: since NP suspensions were filtered using a syringe filter with a pore size of 0.45 μm how much material is lost? Have you quantified the material lost? Based on this data how did you make the loading capacity?
Answer:
Thank you for your careful work. According to our previous study, the vacuum freeze drying method was adopted to quantify the quality of material lost during syringe filter. Briefly, 2 mL of NP suspensions before and after filtered were employed for freeze drying, respectively, and the material lost was the difference in mass. The results showed that the loss of material was extremely small in amount and it can be ignored. In present study, we had not further quantified the material lost. The EE and DL of DTX were calculated as the following equations: EE (%) = [(Wtotal-Wfree)/Wtotal] ×100 % and DL = [(Wtotal-Wfree)/ (Wtotal – Wfree + Wcarrier)] × 100 %, where Wcarrier is the weight of carriers added in the system. To be frank, the omitting of the amount of material lost might underestimate the nanocarrier's loading capacity.
Question 3:
Section 3.3.2: This paragraph is very confusing. G-modification inhibited or enhanced the cell uptake? Because, first it was reported that “G-modification significantly inhibited the uptake of DTXGNPs (1:5) by Caco-2 cells”, but then the opposite such as “G-modification may enhance the cellular uptake of DTX-GNPs via the other bypass pathways such as GLUT, clathrin, micropinocytosis and caveolin”. I think that the authors should be better clarify this concept.
Answer:
Thank you for your instructive suggestions. We have re-writing this paragraph to enhance readability and to clearly convey our findings. 
To further elucidate the cellular uptake mechanism of DTX-NPs and DTX-GNPs by CaCo-2 cells, D-glucose (specific substrate of glucose transporters) and three endocytic inhibitors (chlorpromazine: clathrin pathway inhibitors; amiloride: micropinocytosis pathway inhibitors; indomethacin: caveolin pathway inhibitors) were employed to investigate the cellular uptake mechanism. As shown in Figure. 3C, compared with the control group, the pre-incubation with D-glucose solution hardly affects the uptake of DTX-NPs by CaCo-2 cells, but significantly reduced the uptake of DTX-GNPs, especially in the DTX-GNPs (1:5) group the inhibition rates attained 16.98%. The possible mechanism is that the D-glucose binds to the glucose transporter in pre-incubation phase, resulting in a significant re-duction uptake of DTX-GNPs, suggesting that glucose transporters can mediate the cell uptake of glucose-modified nanoparticles. In addition, compared to control group, addition of chlorpromazine reduced the uptake of DTX-NPs, DTX-GNPs (1:6), DTX-GNPs (1:5) and DTX-GNPs (1:3) by 8.49%, 17.86%, 21.38% and 13.57%, respectively; addition of amiloride reduced the uptake of the above nanoparticles by 15.66%, 19.34%, 22.18% and 19.63%, respectively; addition of indomethacin reduced the uptake of the above nanoparticles by 24.98%, 24.08%, 32.56% and 28.33%, respectively. These results indicated that the caveolin pathway plays an important role in mediating the endocytosis mechanism of nanoparticles. The inhibition endocytosis difference of the four nanoparticles may be due to the modification of glucose affecting the endocytosis pathway of the nanoparticles.
Question 4: 
Section 3.4: Why in your opinion the ka of DTX-GNPs (1:5) was higher than DTX-NPs in the jejunum and ileum?
Answer:
Thanks for your nice comments.
The absorption parameters of various preparation in intestine(x±SD,n=3 )
Preparation    Intestinal segment    Dosage (mg/kg)    Equation    ka /h
DTX solution    Entire intestine    10    y= -0.0598x+5.9626    0.1377±0.0023
DTX-Z NPs    Entire intestine        y= -0.1003x+7.6937     0.2309±0.0019*
    Duodenum        y= -0.2486x+7.2715    0.5725±0.0058
    Jejunum        y= -0.0667x+7.3529    0.1536±0.0011
    Ileum        y= -0.0279x+7.2567    0.0642±0.0037
DTX-GZ-Z(1:5) NPs    Entire intestine        y= -0.1438x+7.9849      0.3311±0.0014* #
    Duodenum        y= -0.1725x+7.6078    0.3972±0.0025#
    Jejunum        y= -0.1501x+7.3868     0.3482±0.0046#
    Ileum        y= -0.1382x+7.3489     0.3156±0.0017#
From the table, we can conclude that the ka values of DTX-GNPs (1:5) group and DTX- NPs group in the whole intestinal segment were 2.40-fold and 1.68-fold of the oral DTX solution group at the same administration dose, respectively, indicating that the two DTX nanocarriers can significantly enhance the intestinal absorption of DTX. Further analysis showed that the ka value of DTX-GNPs (1:5) group in the whole intestinal segment was 1.43-fold of the DTX-NPs group, indicating that glucose modification can further enhance the intestinal absorption of DTX. In the sub-intestinal segments’ exploratory analysis, it was found that the two nanocarriers were absorbed in the duodenum, jejunum, and ileum, respectively, but the DTX-NPs group was mainly absorbed in duodenum, the DTX-GNPs (1:5) group was absorbed uniformly in the three segments. The results of different intestinal segments absorption experiment showed that the ka value of DTX-NPs and DTX-GNPs (1:5) in duodenal was 0.5725 and 0.3972, respectively. The ka value of DTX-GNPs (1:5) in jejunum and ileum was 2.27-fold and 4.92-fold of DTX-NPs, respectively. These results indicated that the modification of glucose significantly changed the intestinal absorption site of DTX-NPs and enhanced the absorption of nanoparticles in jejunum and ileum. According to literatures[1,2], glucose transporters have the most distribution in jejunum, and followed by duodenum. This is consistent with our findings in the present study.
Question 5:
Figure 3C lacks of legend.
Answer:
Thank you for your careful work. We have revised the figure and added the legend. 
Fig. 3. The caco-2 cells uptake and mechanism study of different DTX nanoparticles labeled with C6. (A) The intracellular fluorescence imaging at different time points. (B) Relative cell uptake amount after incubation for different time (mean±S.D., n=3); (C) Inhibition of G and endocytosis inhibitors on the cell uptake after incubation for 3 h (mean±S.D., n=3). *p<0.05 compared with DTX-NPs, #p<0.05 compared with DTX-GNPs (1:5); (D) Absorption rate constants (ka) of different DTX nano-particles in different intestinal segments of rats (*p< 0.05 compared with oral DTX-NPs); (E) Plasma concentration-time profiles in rats after intragastric administration of different DTX nanoparticles (20 mg/kg) (mean±S.D., n=6).
References
1.    Castaneda-Sceppa, C. & Castaneda, F. Sodium-dependent glucose transporter protein as a potential therapeutic target for improving glycemic control in diabetes. Nutr. Rev. 69, 720-729 (2011).
2.    Yamazaki, Y., Harada, S. & Tokuyama, S. Sodium–glucose transporter as a novel therapeutic target in disease. Eur. J. Pharmacol. 822, 25-31 (2018).

Reviewer 2 Report

The authors investigated the oral bioavailability and anti-tumor effect of glucose-modified zein nanoparticles loaded with docetaxel.

The research is interesting and well performed, but some improvements are needed.

 Introduction

1) The author stated that “However, the oral nanocarrier for DTX based on natural proteins such as zein have not been reported”. However, several recent papers have been published on the zein nanoparticles loaded with docetaxel.

Authors should include these references. For example:

-Lee HS, Kang NW, Kim H, et al. Chondroitin sulfate-hybridized zein nanoparticles for tumor-targeted delivery of docetaxel. Carbohydr Polym. 2021;253:117187.

-Wu Z, Li J, Zhang X, et al. Rational Fabrication of Folate-Conjugated Zein/Soy Lecithin/Carboxymethyl Chitosan Core-Shell Nanoparticles for Delivery of Docetaxel. ACS Omega. 2022;7(15):13371-13381.

Statistical analyses

1) Did the authors use a post hoc test for multiple comparison?

 Results

1) To verify the appropriateness of the statistical test used in each graph/table, please indicate in every legend the statistical test used.

2) The author stated “chlorpromazine, amiloride and indomethacin strongly inhibited the cellular uptake of DTX-GNPs (1:5) compared with that of DTX-NPs”, but in fig 3C only indomethacin is statistically significant

 3) As the protein nature of zein can cause immunogenicity in vivo, at least when administered parenterally (Li F, Chen Y, Liu S, et al. The Effect of Size, Dose, and Administration Route on Zein Nanoparticle Immunogenicity in BALB/c Mice. Int J Nanomedicine. 2019;14:9917-9928. Published 2019), have the authors investigated this aspect or are there data in the literature on the immunogenicity of oral zein?

 Minor

1) Several misspellings (i.e. medicine, line 31; lystate, line 186)

2) Write 37°C, instead of 37 ◦C (i.e. lines 178, 180, 183)

Author Response

Reviewer 2:

The authors investigated the oral bioavailability and anti-tumour effect of glucose-mo dified zein nanoparticles loaded with docetaxel.

The research is interesting and well performed, but some improvements are needed.

Question 1:

The author stated that “However, the oral nanocarrier for DTX based on natural proteins such as zein have not been reported”. However, several recent papers have been published on the zein nanoparticles loaded with docetaxel.

Authors should include these references. For example:

-Lee HS, Kang NW, Kim H, et al. Chondroitin sulfate-hybridized zein nanoparticles for tumor-targeted delivery of docetaxel. Carbohydr Polym. 2021; 253:117187.

-Wu Z, Li J, Zhang X, et al. Rational Fabrication of Folate-Conjugated Zein/Soy Lecithin/Carboxymethyl Chitosan Core-Shell Nanoparticles for Delivery of Docetaxel. ACS Omega. 2022;7(15):13371-13381.

Answer:

Thank you for your instructive suggestions. We have added the relative references in the introductory section and edited the description to clearly convey our intentions.

Zein is a hydrophobic plant prolamin obtained from corn that exhibits a helical wheel-like structure consisting of nine homologous repeating units of polypeptides arranged in an anti-parallel manner. Due to high levels of hydrophobic amino acids in the structure, zein is water-insoluble but soluble in more than 50% ethanol solutions. In recent years, zein nanoparticles encapsulating hydrophobic drugs have been demonstrated to increase the bioavailability and treatment effect of water-insoluble drugs, such as rapamycin, resveratrol, quercetin, and docetaxel[1,2]. Although zein NPs were applied to tumor-targeted drug delivery with high drug-loading capacity[1] and controlled-release properties[2], satisfactory intestinal absorption and anti-tumor efficacy could not be attained, as zein NPs displayed poor colloidal stability in biological fluids and limited selectivity to tumor cells[3]. However, recent studies have shown that numerous types of solute carrier transporters, such as glucose and L-amino acids, which are present throughout the gastrointestinal tract and tumor cell membranes[4], may represent potential targeting sites for successful oral delivery.

Question 2:  

Did the authors use a post hoc test for multiple comparison?

Answer:

Thank you for your valuable advice. In original manuscript, we had not performed the post hoc test for multiple comparison. As per your suggestions, we have performed the post hoc test to further explore the sample significance level using the Bonferroni method. Some of the statistical results are described below:

Time points at 2 h.

Time points at 4 h.

Compared with DTX-GNPs (1:3) group, the other groups showed a lower uptake by CaCo-2 cells at 1 h, and the difference was statistically significant (p<0.05). Among the other three groups, there are no statistically significant difference (p>0.05).

Compared between any two groups, the intra-groups showed statistically significant difference (p<0.05) in cellular uptake of CaCo-2 cells at 2 h and 4 h. Further mean difference analysis revealed that the DTX-GNPs (1:5) group exhibited a stronger cellular uptake capacity by CaCo-2 cells compared to the other groups.

We also performed the post hoc test for multiple comparison for any datum which the statistical test needs to analyze of variance. The results of the post hoc test for multiple comparison are consistent with the conclusion that we had made.

Thank you again for your guiding advice. In the future scientific research work, we will conduct more standardized statistical processing and then draw the conclusions.

Question 3:

To verify the appropriateness of the statistical test used in each graph/table, please indicate in every legend the statistical test used.

Answer:

Thank you for your valuable advice. We have added the statistical test methods for the graph/table.

Question 4:

The author stated “chlorpromazine, amiloride and indomethacin strongly inhibited the cellular uptake of DTX-GNPs (1:5) compared with that of DTX-NPs”, but in fig 3C only indomethacin is statistically significant.

Answer:

Thank you for your careful work. We have revised the figure and re-writing the section of 3.3.2 “Cellular uptake mechanism”. We also re-analyzed the raw data and the results showed that the addition of chlorpromazine reduced the uptake of DTX-NPs, DTX-GNPs (1:6), DTX-GNPs (1:5) and DTX-GNPs (1:3) by 8.49%, 17.86%, 21.38% and 13.57%, respectively; the addition of amiloride reduced the uptake of the above nanoparticles by 15.66%, 19.34%, 22.18% and 19.63%, respectively; the addition of indomethacin reduced the uptake of the above nanoparticles by 24.98%, 24.08%, 32.56% and 28.33%, respectively. These results indicated that the caveolin pathway plays an important role in mediating the endocytosis mechanism of nanoparticles. The inhibition endocytosis difference of the four nanoparticles may be due to the modification of glucose affecting the endocytosis pathway of the nanoparticles.

Question 5:

As the protein nature of zein can cause immunogenicity in vivo, at least when administered parenterally (Li F, Chen Y, Liu S, et al. The Effect of Size, Dose, and Administration Route on Zein Nanoparticle Immunogenicity in BALB/c Mice. Int J Nanomedicine. 2019; 14:9917-9928. Published 2019), have the authors investigated this aspect or are there data in the literature on the immunogenicity of oral zein?

Answer:

Thank you for your valuable advice. In the present manuscript, we have not investigated the immunogenicity of zein when administered gastrointestinal route, and the potential mechanism why zein can cause immunogenicity. But we are very inspired by this comment, we have systematically searched the literature database such as PubMed, Web of Science and etc. To date, only limited information is available concerning zein immunogenicity, and some contrary results have been reported. Islam et al[5] have investigated the oral immunogenicity of zein or zein nanoparticles in mice, and the results demonstrate that zein nanoparticles did not show any immunogenicity in mice. However, Pepi et al[6] have drew a contrary conclusion that zein microspheres were immunogenic when administered intramuscularly and orally in mice.

Minor

1) Several misspellings (i.e. medicine, line 31; lystate, line 186)

2) Write 37°C, instead of 37 ◦C (i.e. lines 178, 180, 183)

Answer:

Thank you for your careful work. We have revised the misspellings and polished the manuscript by a professional assistant in writing.

References

  1. Lee, H.S., et al. Chondroitin sulfate-hybridized zein nanoparticles for tumor-targeted delivery of docetaxel. Carbohydr. Polym. 253, 117187 (2021).
  2. Wu, Z., et al. Rational Fabrication of Folate-Conjugated Zein/Soy Lecithin/Carboxymethyl Chitosan Core-Shell Nanoparticles for Delivery of Docetaxel. ACS Omega 7, 13371-13381 (2022).
  3. Sun, C.X., Chen, S., Dai, L. & Gao, Y.X. Structural characterization and formation mechanism of zein-propylene glycol alginate binary complex induced by calcium ions. Food Research International 100, 57-68 (2017).
  4. Wu, L., et al. Bioinspired butyrate-functionalized nanovehicles for targeted oral delivery of biomacromolecular drugs. J Control Release 262, 273-283 (2017).
  5. Islam, M.S., et al. Bioadhesive Food Protein Nanoparticles as Pediatric Oral Drug Delivery System. ACS Appl Mater Interfaces 11, 18062-18073 (2019).
  6. Hurtado-Lopez, P. & Murdan, S. An investigation into the adjuvanticity and immunogenicity of zein microspheres being researched as drug and vaccine carriers. J Pharm Pharmacol 58, 769-774 (2006).

Reviewer 3 Report

The manuscript is interesting and in line with journal aims. Before publishing, some point must be revised:

CaCo-2 is the correct form.

The IR data are confusing due to image low quality. Please improve.

An image of the nanoparticles must be included.

The slots in figure 3 are very low, please revise.

Also, the statistycal analysis must be revised because not clear, why n=3 or n=6. please uniform it.

the PK data in table 2 are very difficult to understand because not supported by a good discussion. please improve it.

figure 4B is not clear, please increase its size.

Author Response

Reviewer 3:

The manuscript is interesting and in line with journal aims. Before publishing, some point must be revised:

Question 1:

CaCo-2 is the correct form.

Answer:

Thank you for your careful work. As per your valuable advice. We have revised the writing format.

Question 2:  

The IR data are confusing due to image low quality. Please improve.

Answer:

Thank you for your guiding advice. The original figures of the IR spectrum have been added as the attachment.

Figure 1. The characterization of GZ synthesized. (A) The FTIR spectroscopy curve. (B) The 1H-NMR spectrum of Z and GZ. (C) The isoelectric point of Z and GZ (mean ± S.D., n=3).

Question 3:

An image of the nanoparticles must be included.

Answer:

Thank you for your guiding advice. The TEM image of DTX-NPs and DTX-GNPs has been added. We have revised the Figure 2 in the main text.

Fig 2. In vitro characterization of DTX-GNPs (1:5). (A) The distribution of DLS particle size. (B) Zeta potential distribution. (C) The TEM image of DTX-NPs. (D) The TEM image of DTX-GNPs. (E) In vitro release in pH 1.2 HCl solution for the first 2 h and then in pH 6.8 PBS solution (mean±S.D., n=3).

Question 4:

The slots in figure 3 are very low, please revise.

Answer:

Thank you for your careful work. As per your valuable advice. We have revised figure 3 for enhanced formality and readability.

Fig. 3. The caco-2 cells uptake and mechanism study of different DTX nanoparticles labeled with C6. (A) The intracellular fluorescence imaging at different time points. (B) Relative cell uptake amount after incubation for different time (mean±S.D., n=3); (C) Inhibition of G and endocytosis inhibitors on the cell uptake after incubation for 3 h (mean±S.D., n=3). *p<0.05 compared with DTX-NPs, # p<0.05 compared with DTX-GNPs (1:5); (D) Absorption rate constants (ka) of different DTX nano-particles in different intestinal segments of rats (*p < 0.05 compared with oral DTX-NPs); (E) Plasma concentration-time profiles in rats after intragastric administration of different DTX nanoparticles (20 mg/kg) (mean±S.D., n=6).

Question 5:

Also, the statistical analysis must be revised because not clear, why n=3 or n=6. please uniform it.

Answer:

Thank you for your careful work. We have checked the statistical analysis and added the statistical test method to ensure that each graph/table can clearly convey our findings and exhibit the main results. But unfortunately, we did not do better. We have made some detailed explanations for why n=3 or n=6 in the graph/table.

In the graph/table (Table 1, Fig 2D, Fig 3B, Fig 3C and Fig 4B), where n=3 usually means that the samples were measured three times or the experiments were repeated three times, and the results were expressed as the mean ± standard deviation; But in the “Pharmacokinetic studies” and “Biodistribution and anti-tumor effect” sections, where n=6 or n=8 means that the number of mice or rats were included in each experiment group.

Question 6:

The PK data in table 2 are very difficult to understand because not supported by a good discussion. please improve it.

Answer:

Thank you for your guiding advice. We have added the relative discussion and re-writing the description in the main text. From table 2, the oral bioavailability of DTX-NPs group and DTX-GNPs (1:5) group was 43.82% and 96.04%, respectively. Compared with the DTX-NPs group, the modification of glucose significantly enhanced the oral absorption and the bioavailability of DTX-NPs, and the absolute bioavailability of DTX-GNPs (1:5) was 2.19 times higher than DTX-NPs. Similarly, the Tmax of the DTX-GNPs (1:5) was extended to 2 h and the MRT increased 8.07 h, and the AUC0-24 h was 2.19 times greater than DTX-NPs. In addition, among the different preparation groups, the oral bioavailability was also varied from the grafting degree of G. Among them, the absolute oral bioavailability of DTX-GNPs (1:5) was 2.86, 1.54 and 1.95 times higher than DTX-GNPs (1:0), DTX-GNPs (1:3) and DTX-GNPs (1:6), respectively, indicated that the glucose density on the surface of the nanoparticles directly affected the oral bioavailability, and showed the trend of first rising and then decreasing with the increase of glucose density.

Question 7:

Figure 4B is not clear, please increase its size.

Answer:

Thank you for your guiding advice. The original figure of Figure.4 has been added as follows:

Fig. 4. Ex vivo distribution of different nanoparticles labeled with IR-780 iodide after intragastric administration in tumor-bearing mice. (A) Fluorescent imaging. (B) Quantitative analysis based on fluorescence intensity (mean±S.D., n=3).

Round 2

Reviewer 1 Report

The authors properly revised the manuscript, so it can be accepted in the present form.